# Relative Neuroadaptive Effect of Resistance Training along the Descending Neuroaxis in Older Adults

**DOI:** 10.3390/brainsci13040679

**Published:** 2023-04-18

**Authors:** Mattias Romare, Guilherme H. Elcadi, Elin Johansson, Panagiotis Tsaklis

**Affiliations:** 1ErgoMech-Lab, Department of Physical Education and Sport Science, University of Thessaly, 42100 Trikala, Greece; 2Division of Ergonomics, School of Engineering Sciences in Chemistry, Biotechnology and Health, KTH Royal Institute of Technology, 14157 Huddinge, Sweden; 3Pain in Motion Research Group, Departments of Human Physiology and Rehabilitation Sciences, Faculty of Physical Education and Physiotherapy, Vrije Universiteit Brussel, PC 1050 Brussel, Belgium; 4Centre of Orthopaedics and Regenerative Medicine, C.O.R.E.-C.I.R.I., Aristotle University of Thessaloniki, 541 24 Thessaloniki, Greece; 5Department of Molecular Medicine and Surgery, Karolinska Institute, SE-171 76 Solna, Sweden

**Keywords:** aging, neural adaptations, resistance training, physical function

## Abstract

Age-related decline in voluntary force production represents one of the main contributors to the onset of physical disability in older adults and is argued to stem from adverse musculoskeletal alterations and changes along the descending neuroaxis. The neural contribution of the above is possibly indicated by disproportionate losses in voluntary activation (VA) compared to muscle mass. For young adults, resistance training (RT) induces muscular and neural adaptations over several levels of the central nervous system, contributing to increased physical performance. However, less is known about the relative neuroadaptive contribution of RT in older adults. The aim of this review was to outline the current state of the literature regarding where and to what extent neural adaptations occur along the descending neuroaxis in response to RT in older adults. We performed a literature search in PubMed, Google Scholar and Scopus. A total of 63 articles met the primary inclusion criteria and following quality analysis (PEDro) 23 articles were included. Overall, neuroadaptations in older adults seemingly favor top-down adaptations, where the preceding changes of neural drive from superior levels affect the neural output of lower levels, following RT. Moreover, older adults appear more predisposed to neural rather than morphological adaptations compared to young adults, a potentially important implication for the improved maintenance of neuromuscular function during aging.

## 1. Introduction

The origins of age-related decline in maximal voluntary force production could be said to stem from a combination of adverse musculoskeletal alterations and detrimental changes along the descending neuroaxis [1,2,3]. As early as the 1990s, age-related loss in neuromuscular function was argued to be one of the main contributors to the onset of physical disability in older adults [3]. From a public health perspective, older adults have been shown to be more susceptible to fall-related injuries, likely due to losses in neuromuscular functioning with aging [4]. Moreover, longitudinal studies have demonstrated that annual losses of voluntary muscle activation are 2–5 times greater than those of skeletal muscle mass [2,5,6], and that age-related strength losses are only modestly associated with the loss of skeletal muscle mass [7,8,9]. This may indicate a neurological origin in the onset of age-related strength and function losses. More recently, Manini and Clark [2] suggested the use of the term “dynapenia” to describe the temporal loss of neuromuscular functioning throughout the aging process [2,3,10]. The model, in addition to morphological and peripheral factors, takes into consideration the impact of decreased corticospinal excitability along different levels of the descending neuroaxis, contributing to decreased availability of descending neural drive carried to the muscle [2,10]. Hence, this highlights the importance of adequate functioning throughout all levels of the descending neuroaxis to recruit motor-neuron pools voluntarily.

Importantly, within the discourse of age-related decline of neuromuscular functioning, it is critical to separate the intrinsic contractile properties of a motor unit (MU) and the voluntary ability to maximally recruit the motor-neuron pool. As noted by Clark and co-workers [2,10], the term muscle strength is a common misnomer, as it implies the intrinsic force generating capacity of the muscle in the absence of endogenous descending neural drive to the muscle. The latter may be evaluated through exogenous supramaximal electrical stimulation of the muscle, where the resulting electromyographical (EMG) amplitude, known as the M-max, represents the inherent contractile potential of the muscle [11]. The comparison of M-max with the EMG amplitude of a maximal voluntary isometric contraction (MVIC) can evaluate the ability of descending corticospinal pathways to maximally recruit their MU pool, and is termed VA. Moreover, without normalization to M-max, no meaningful determination of VA can be made [12,13]. The interpolated twitch technique (ITT) is often used to evaluate deficits in VA, as first described by Merton [12]. ITT utilizes electrical stimulation of a peripheral nerve at a plateaued peak amplitude of an MVIC and quantifies the additional generated increase in surface electromyography (sEMG) amplitude. The discrepancy in voluntary and involuntary activation is commonly termed neural strength deficit, expressed as a percentage of VA normalized to M-max; see Equation (1) [13].
(1)$$VA\left(\%\right)=\left(1-[\frac{Interpolated\;twitch\;amplitude}{Resting\;control\;twitch\;amplitude}]\right)\times 100$$

Equation (1). Calculation of the voluntary activation using the interpolated twitch technique. A higher VA (%) indicates a greater ability of the central nervous system to activate the muscle voluntarily. Abbreviations: VA, voluntary activation, MVIC, maximum voluntary isometric contraction [13].

Alternatively, a similar method can be used by evaluating the added force in the presence of superimposed electrical stimuli during MVIC plateau, known as the central activation ratio (CAR). It is expressed as a percentage of maximum isometric force produced in the presence of a superimposed electrical stimulus; see Equation (2) [13].
(2)$$CAR=\left(\frac{MVIC\;plateau\;force+Superimposed\;force}{MVIC\;plateau\;force}\right)$$

Equation (2). Calculation of central activation ratio. Improved CAR indicates a greater ability of the central nervous system to activate the muscle voluntarily. Abbreviations: CAR, central activation ratio, MVIC, maximum voluntary isometric contraction [13].

Furthermore, several techniques can provide insights in the functioning of the descending levels of the central nervous system. These techniques, when used in combination with the abovementioned methods, may discern the relative contribution of each level to VA. At the corticospinal level, excitability can be evaluated by techniques such as transcranial magnetic stimulation (TMS), a non-invasive technique that generates an oscillating magnetic field directed toward the primary motor cortex (M1) or toward the cervicomedullar region eliciting non-voluntary motor-evoked potentials (MEP) and can be recorded by sEMG [14]. At the spinal level, excitability may be measured by techniques such as H-reflex, which evaluates the excitability of monosynaptic reflex arches. The latter is performed by the antidromical electrical stimulation of the peripheral nerves, sent through Ia afferents [15,16,17]. This produces an excitation in the spine and a monosynaptic response, recorded by sEMG [15]. In addition, volitional waves (V-wave), as proposed by Upton and colleagues [18], can be used to determine the net descending corticospinal drive. The technique involves performing an MVIC at the instance of an amplitude cancelation of the H-wave, caused by antidromical collision. If the added supraspinal drive overcomes the antidromical collision, the H-wave re-appears and is denoted a V-wave [19]. The reappearance signifies added descending neural drive from cortical areas, adding to the net descending neural drive from corticospinal levels.

Lastly, at the furthest periphery of the descending neuroaxis, motor-unit discharge rates (MUDRs) and arguably motor-unit synchronization may also play a role in the ability to voluntarily activate muscles, as it has been shown that incremental force production during a voluntary contraction requires increased MUDR, in addition to increased motor-unit activity and recruitment of additional motor units [20,21,22]. However, the impact of MU synchronization on VA has been recently brought into question [23]. Furthermore, given that the descending neural drive to the muscle precedes the onset of skeletal muscle contraction, it may be plausible that any preceding impairments along the descending neuroaxis impairs the subsequent activation of skeletal muscle and VA [24,25,26,27,28,29]. Hence, decreases in the descending neural drive could disallow activation of HT-MUs [30,31,32,33,34], thereby inducing losses of HT-MUs in older adults, increasing the risk of falls and decreasing physical performance [2,27,28,29]. The latter underlines the importance of preserving adequate functioning throughout all levels of the descending neuroaxis during the aging process to preserve physical performance and strength. Thus, identifying viable and safe modalities to increase VA in older adults is of great importance.

Resistance training (RT) is an effective and practical way to offset age-related muscle loss and improve strength [35,36,37]. It is generally safe and well tolerated by the elderly population [8,35,36,37]. RT may be defined as a training method where skeletal muscle contractions are performed against external resistance in a progressive manner over time. The general goal of RT is to increase skeletal muscle mass, strength, and endurance. Commonly, RT can be performed using one’s bodyweight, free-weights, or machines to supply adequate stimuli for neuromuscular adaptations. It is known that in healthy young adults, RT elicits adaptations of several levels of the descending neuroaxis, improving VA. Among the adaptations are increases in net corticospinal excitability, in addition to improved maximal MUDR [19,35,36,37,38]. Recently, it has been suggested that increased strength following RT in young and older adults may be due to different adaptation mechanisms, being mainly muscular in young adults and mainly neural in older adults [39]. Accordingly, it has been shown previously that older adults display decreased resting excitability for several levels compared to young adults [2,10,40]. However, less is known about the relative neuroadaptive contribution of RT in older adults. Given that the rate of age-related voluntary muscle activation loss exceeds those of skeletal muscle mass, and that neural drive to the muscle precedes skeletal muscle contractions, it may be of clinical value to outline where along the descending neuroaxis such neural adaptation in response to RT occurs. This may aid the development of effective training strategies to combat the age-related decline in neuromuscular function and improve physical autonomy. Thus, the aim of the present review is to outline the current state of the literature, regarding where and to what extent neural adaptations occur along the descending neuroaxis in response to resistance training in older adults. Emphasis was placed upon the relative neuroadaptive contribution of each level of VA.

## 2. Materials and Methods

The present review was conducted in accordance with the up-to-date Preferred Items for Systematic Reviews and Meta-Analyses (PRISMA) guidelines [41]. The PEDro scale, consisting of 11 criteria covering external validity (criterion 1), internal validity (criterion 2–9), and statistical reporting (criterion 10–11) was used to determine the external and internal validity of the studies [42]. For this review, criteria 1, 2, 4 and 8–11 were used (See Table A1, Appendix A). The validation criteria 5–7 according to the original PEDro scale were excluded, as these points addressed blinding procedures, which are commonly not viable due to the nature of RT studies. For this review, with above points excluded, a maximum score of 7 was determined. Based on this, we considered articles scoring 6–7 points as ‘excellent’ quality, articles scoring 4 and 5 points were deemed ‘moderate’ and ‘good’ quality, respectively, and scores of <4 points were determined to be of ‘poor’ quality. Articles scoring lower than 4 points were systematically excluded from the present review. Results of the PEDro scale evaluation can be seen in Table A2, Appendix A and the complete search matrix is included online within the supplementary uploaded material of this review.

### 2.1. Search Strategy: Databases and Eligibility Criteria

The initial step was to conceptualize the outline of the review and to define the research question. Subsequently, a dynamic search strategy was formulated, and a literature search was conducted using the search engines PubMed, Google Scholar and Scopus. The initial pre-selection of the papers was performed in July of 2022 by reading the titles and abstracts. Following this preliminary exclusion phase, the remaining articles were read and selected based on the level of suitability for inclusion in relation to the research question by two independent reviewers, and in relation to the pre-determined PEDro quality assessment criteria (See Table A2, Appendix A). A critical approach was used for the result synthesis of the overarching questions of the review, and tables of summary were created from the papers that were deemed of acceptable quality (>4 score).

### 2.2. Inclusion and Exclusion Criteria

The scope of this review is limited to primarily interventional and secondarily cross-sectional studies evaluating the effects of RT on neural adaptations and voluntary muscle activation in older adults. Older adults were, for the purpose of this review, defined as being >65 years of age, as previously defined by Witham [43]. Animal studies and studies featuring only young adults were excluded, as were studies featuring neuromuscular pathologies that may affect corticospinal excitability. Studies featuring older adults were found relevant if containing the following features based on the descending neuroaxis: (I) cortical-level studies evaluating supraspinal adaptations using TMS following RT in older adults, (II) corticospinal-level studies evaluating corticospinal adaptations using TMS and V-wave methodologies following RT in older adults, (III) resistance training studies evaluating EMG parameters following RT in older adults, (IV) peripheral excitability studies evaluating motor-unit discharge rates and synchronization following RT in older adults, and (V) studies evaluating voluntary muscle activation using ITT or CAR separately or in combination with the methodologies of I–IV, following RT in older adults.

### 2.3. Literature Search

A literature search was initially carried out in PubMed, Google Scholar and Scopus. A summary of the search strategy can be seen in Figure 1. Search keywords were used alone or in combination to generate results that met the criteria for the present review. Only studies in the English language and published within the last 50 years were accepted. The searches that were performed for this review included both MeSH terms and keywords that contributed more than two papers to the search. Identical keywords were used for all databases with the appropriate Boolean operations (e.g., AND, OR). The MeSH terms, keywords and their corresponding Boolean operations used in the search are listed, in no particular order, below:

Sarcopenia, dynapenia, voluntary muscle activation, neural adaptations, neuroplasticity, interpolated twitch technique, resistance training, muscle strengthening exercise, transcranial magnetic stimulation, TMS, h-reflex, h-wave, v-wave, SICI, LICI, motor-unit rate coding, descending neural drive, skeletal muscle mass, motor-evoked potentials, resting motor threshold, silent period, maximum voluntary isometric contraction, central activation ratio, and cognition.

Transcranial magnetic stimulation AND resistance training AND older adults)) OR (spinal excitability AND resistance training AND older adults)) OR (resistance training AND older adults AND intracortical inhibition)) OR (resistance training AND older adults AND SICI AND LICI)) OR (resistance training AND older adults AND Hoffman reflex)) OR (resistance training AND older adults AND cortical silent period)) OR (resistance training AND older adults AND v-wave)) OR (resistance training AND older adults AND presynaptic inhibition)) OR (resistance training AND older adults AND rate coding)) OR (resistance training AND older adults AND sEMG coherence)) OR (resistance training AND older adults AND rheobase)) OR (resistance training AND older adults AND chronaxie)) OR (resistance training AND older adults AND strength-duration)) OR (resistance training AND older adults AND interpolated twitch)) OR (resistance training AND older adults AND voluntary activation)) OR (resistance training AND older adults AND central activation ratio).

## 3. Results

Initially 317 articles were identified in the primary PubMed and Google Scholar search by scrutinizing title, abstract and conclusion. In total, 63 articles met the initial inclusion criteria for a follow-up quality assessment. Next, 40 articles were excluded, due to featuring neuropathologies, only non-older adult populations and animal studies. Following the primary screening process, a secondary PEDro quality screening process was implemented by scrutinizing the remaining 23 articles in their entirety, which were deemed of adequate quality to be included within the current review. The PEDro quality scores for the 23 studies ranged from ‘good’ to ‘excellent’, out of which 22 of the included articles scored ‘excellent’ and one article scored ‘good’, displaying very similar qualities, and ranging between 5 and 7 points. Overall, the included studies displayed low methodological heterogeneity in the evaluation of features I–V. The included studies were compiled into tables based on the included study’s main outcome parameters, as described above in the inclusion criteria (I–V). A flowchart model of the literature search is shown below, in Figure 1.

### 3.1. Studies Evaluating Supraspinal Adaptations in Older Adults Following Resistance Training

Cortical areas represent one of the first levels of investigation for neuroadaptations along the descending neuroaxis, following RT. Given that the ensemble of volitional actions such as movement originate from areas such as the cerebellum and primary motor-cortex (M1), these areas may represent potential sites of interests for RT-induced neuroadaptations [44]. Descending neural drive is to a large part dependent on disinhibition and/or increased corticomotor excitability, primarily in the M1 region [14,45]. Recent developments and accessibility of transcranial magnetic stimulators (TMS) have made in vivo investigations of supraspinal adaptations in response to RT more readily available. In addition, it is possible to discriminate between relative neuronal adaptations of the motor cortex and the spine separately using different stimulation protocols like single or paired pulses. TMS is a non-invasive technique that generates oscillating magnetic fields using a Hodgkin–Huxley figure coil directed towards areas such as the M1, eliciting MEPs by electromagnetic induction [46], and it is recorded with surface electromyography (sEMG).

We identified four studies evaluating supraspinal adaptations in older adults following RT, using mainly TMS methodologies (See Table 1) [47,48,49]. Penzer and co-workers [48] demonstrated greater MVIC increases following RT compared to balance training in older adults. Increased MVIC coincided with increased CAR and reduced by 50% the H-max threshold (*p* = 0.036). However, the M-max and maximal MEP amplitude remained unchanged in both groups (*p* > 0.05). Penzer argued that the above indicates that RT increases maximal strength and changes the neural control of lower-leg muscles during upright standing [48]. Next, Kamen and co-workers [47] reported that neither M-max nor MEP max amplitude were changed after RT for either group (*p* = 0.42, *p* = 0.23, respectively). However, strength gains were positively correlated with the increase in CAR (*p* < 0.001). Conversely, cortical silent period (CSP) duration was reduced (*p* < 0.001) in parallel with increased MVIC for both young and older adults (*p* < 0.001), but to a larger extent for older adults (*p* < 0.05) [47]. However, Kamen and co-workers did not detect changes in the MEP max (*p* = 0.69) or H-max (*p* = 0.38), coinciding with MVIC for older adults, arguing that corticomotor disinhibition may explain these results. Lastly, Otieno and co-workers [49] evaluated the effect of a single bout of low-load isometric RT. The results of the latter study showed no change in SICI duration following isometric RT in either group (*p* > 0.05). However, only older adults acutely decreased LICI following RT (*p* < 0.05), while young adults did not display any changes (*p* > 0.05). The results of these studies showed overall reductions in cortical inhibition, such as CSP and LICI duration, in parallel with increased MVIC in older adults compared to young, following RT [47,48].

### 3.2. Studies Evaluating Corticospinal Adaptations in Older Adults Following Resistance Training

The next possible site of investigation along the descending neuroaxis may be the corticospinal tract. Commonly, corticospinal tract excitability is evaluated using methods such as the Hoffman reflex and V-wave technique. Specifically, H-waves provide indirect estimations in the net excitability of spinal areas, stemming from the intrinsic excitability of the monosynaptic reflex arches, inhibitory functioning of the Ia afferents and arguably the Renshaw cells’ function [50]. The H-reflex technique involves the administration of the submaximal antidromical electrical stimulation of specific peripheral nerves in resting conditions. Similarly, V-waves are recorded during a voluntary contraction following the disappearance of the H-wave due to amplitude cancelation, and can be interpreted as the ability of the descending neural drive to overcome the state of amplitude cancelation. Therefore, V-wave amplitude is said to be a measure of total corticospinal excitability [25,26,51].

We identified four studies that evaluated the main outcomes for corticospinal excitation (H- and V-waves) following RT in older adults (See Table 2). Two studies by Unhjem and co-workers evaluated the effect of RT on normalized H-wave and V-wave amplitude (H/M and V/M-ratio) [52,53]. The authors [52,53] demonstrated that older adults showed decreased V/M ratios (*p* < 0.05) and prolonged H-wave latencies (*p* < 0.05) at baseline, but no differences for H/M ratios compared to young (*p* > 0.05). Following RT of the knee extensors, V/M ratio and MVIC showed greater increases in older adults compared to young (*p* < 0.05), but these were not accompanied by improved H/M ratio (*p* > 0.05) [52]. In a later study, Unhjem and co-workers [53] showed greater V/M-max ratio improvements in the plantar flexors in older adults following RT compared to young (*p* = 0.01), but no change in H/M-ratio for either group (*p* > 0.05). Similarly, Tøjen and co-workers [54] noted parallel increases in MVIC (*p* < 0.05) and V/M ratio (*p* < 0.05) in the absence of changes in the H/M-ratio in older adults compared to young (*p* > 0.05). Notably, all the studies that featured secondary outcome measurements of VA displayed significant improvements in VA alongside increased V/M ratio [52,54,55]. Additionally, Tøjen and co-workers found that parallel increases in MVIC and VA were correlated (*p* < 0.05). Interestingly, however, Scaglione and co-workers [55] did not use V-wave methodologies, and the authors argued that increases in VA in the absence of H-wave amplitude were due to either increased corticomotor excitation or improved peripheral excitability. The latter was depicted in the decreased contraction time (CT) following RT (*p* < 0.05). Thus, the studies showed decreased V-wave amplitude at baseline and improvements in maximal V-wave amplitude and MVIC following RT in older adults compared to young [52,53,54,55]. Moreover, three out of the four studies showed no improvement in H-max following RT, indicating a lack of change in spinal excitability.

### 3.3. Studies Evaluating EMG Parameters in Older Adults Following Resistance Training

As action potentials descend to the muscle from the corticospinal tract, VA will be largely influenced by the progressive addition of MUs in accordance with the Henneman size principle [21,25,26]. During incremental force production, LT-MU are generally recruited first, since they require less neural excitation [20]. To further increase force production, there is a progressive recruitment of the motor neuron pool or the selective recruitment of HT-MU, which requires a greater descending neural drive [21,56]. Thereby, it has been argued that the sum of the myoelectrical activity to the muscle and the MU depolarization is strongly associated with the descending neural drive to the motor neuron pool [21]. Therefore, EMG is commonly used to quantify and describe muscle activation patterns. Surface electrodes (sEMG), intramuscular needle electrodes or high-density surface electrode grids can measure EMG. However, due to the non-invasive nature of sEMG, it is often preferred for evaluating training adaptations in regard to muscle activity, behavior and neural drive [20].

We identified five studies evaluating the main outcomes of sEMG parameters following RT in older adults (See Table 3). One study by Suetta and co-workers [57] compared traditional rehabilitation training and electrical muscle stimulation (EMS) strategies to RT in older adults following hip-replacement surgery. The authors found a greater increase in RMS amplitude and in the evoked rate of twitch force development (RFD) and impulse velocity following the fifth week in the RT group, compared to control conditions. In addition, Macaluso and co-workers [58] noted differences in “early” and “late” sEMG root-mean-square (RMS) and mean power frequency (MPF) responses between older and young women following 6 weeks of RT. The “early” RMS and MPF were defined as the averaged data points at 3–6 s of a 12-second MVIC, while “late” RMS and MPF were defined as seconds 9–12 of the same total duration. Macaluso and co-workers noted that “early” RMS amplitude increased for younger women (*p* < 0.05), but not for older women after RT. Conversely, “late” RMS amplitude increased for older women (*p* < 0.05), but not for young women (*p* > 0.05) after RT [58]. The latter study was the only study evaluating changes in MPF and did not detect any significant changes in either group [58]. Lastly, Häkkinen and co-workers [59] and De Boer and co-workers [60] evaluated the effect of long-term RT (6–12 months) on agonist–antagonist co-activation RMS amplitude during plantar (PF) and dorsiflexion (DF) MVIC in older adults. The authors also evaluated MVIC and concentric 1RM performance. De Boer and co-workers [60] noted that the RT group increased PF MVIC for all ankle joint angles (*p* < 0.05), but showed a decreased DF MVIC (*p* < 0.05). Furthermore, it was reported that PF RMS amplitude increased (*p* < 0.05) and DF co-activation decreased (i.e., antagonist activation) for all joint angles (*p* < 0.05), while no changes were observed in the control group [60]. In addition, Häkkinen and co-workers noted a higher increase in MVIC for older adults compared to middle-aged controls (*p* < 0.001) and a decrease in antagonist co-activation during an MVIC of the knee extensors following the first two months of training (*p* < 0.05). All studies demonstrated significant increases in MVIC following RT [57,58,59,60], with the exception of Cannon and co-workers [61], who did not detect any differences in MVIC increases for young and older women following RT. Furthermore, for all the studies, an increased MVIC following RT was mirrored by parallel increases in the EMG RMS amplitude [57,58,59,60,61].

### 3.4. Studies Evaluating Peripheral Excitability and Motor-Unit Discharge Rates in Older Adults Following Resistance Training

While muscle force is largely dependent on its cross-sectional area, the central nervous system can regulate force production not only by the recruitment of additional motor units [25] but also through altering intrinsic MU-firing behaviors, stemming from adaptations at the level of the excitation–contraction coupling (ECC). This level could be viewed as the final common denominator of neural adaptations along the descending neuroaxis [62]. At this level, muscle contractions are achieved by the interaction between the descending neural drive and voltage-sensitive dihydropyridine receptors (DHP) in the t-tubular system. This interaction initiates a release of calcium ions (Ca^2+^) from the sarcoplasmic reticulum through ryanodine gates and sarcoendoplasmic reticulum calcium ATPase (SERCA) 1–2 pumps, allowing the formation of actin–myosin bonds [63]. While there is a strong relationship between the number of MUs recruited and the force production, optimal force production can also be regulated by changing the rate at which they discharge [64,65]. Motor-unit discharge rate (MUDR) is thus dependent on factors that influence action potential behaviors.

We identified four studies that evaluated peripheral excitability and motor-unit discharge rates in older adults, following RT (See Table 4). One study, by Patten and co-workers [66], evaluated the effect of a 6-week isometric RT protocol of the abductor minimi muscle. At baseline, maximal MUDR was lower for older adults compared to young (*p* < 0.001). RT increased maximal MUDR and MVIC for both young and older adults (*p* < 0.05), but to a greater degree for older adults compared to young (*p* < 0.05). However, during the second week of RT, maximal MUDR returned to baseline, which coincided with a cross-over education in the untrained limb, shown by increased MVIC and maximal MUDR only in the older-adults group (*p* < 0.05). Subsequently, a study by Griffin and co-workers [67] evaluated the effect of 4 weeks of low-load RT training of the adductor pollicis and first dorsal interosseous muscle on MU synchronization and MUDR variability during a sustained contraction at 2, 4, 8, and 12% MVIC. Following RT, only older adults improved force variability for both muscles at 2 and 4% of MVIC (*p* < 0.001), and no improvements were seen at 8 and 12% MVIC. Moreover, only older adults decreased maximal MUDR for both muscles at 2, 4 and 8% MVIC (*p* < 0.01), while MU synchronization did not improve for either group following RT (*p* > 0.05). In addition, a study by Kamen and co-workers [68] also showed lowered baseline maximal MUDR (*p* < 0.05) and greater increased maximal MUDR at 100% MVIC (*p* < 0.05) for older adults compared to young, following RT. However, no improvements in maximal MUDR were found at 10 and 50% MVIC for any of the groups. Lastly, Knight and co-workers [69] evaluated maximal MUDR and VA following a 6-week RT protocol. At baseline, and following RT, both groups showed significant associations between maximal MUDR and MVIC torque (r = 0.79, *p* < 0.05). Moreover, for older adults, a significant association between maximal MUDR and CAR was found (r = 0.72, *p* < 0.05). Following RT, both groups similarly improved MVIC torque and maximal MUDR (*p* < 0.05) but with no group differences (*p* > 0.05). For both groups there was a significant association between maximal MUDR and decreased ITT amplitude (r = 0.62, *p* < 0.05).

### 3.5. Voluntary Muscle Activation Studies in Older Adults Following Resistance Training

Techniques such as the ITT and CAR have been used to evaluate the discrepancy between the involuntary maximal contractile potential of a muscle by electrical stimulation and the ability to voluntarily activate a muscle, thus providing a normalized metric for VA. These techniques also provide the opportunity to evaluate the effect of RT on VA along the descending neuroaxis. When used in combination with previously discussed techniques, the relative neuroadaptive contribution of each level of the descending neuroaxis on VA can be understood.

We identified six studies that combined methodologies that primarily evaluated adaptations of the descending neuroaxis in conjunction with VA protocols (See Table 5). All the identified studies quantified VA but none used parallel measures of corticospinal excitability (e.g., V-waves and TMS). The majority of the studies reported baseline reductions in VA for older adults compared to young adults [39,69,70,71,72]. However, Knight and co-workers [73] reported that VA was 2% lower for older adults, but not significant (*p* > 0.05). Conversely, Walker and Häkkinen [39] reported that older adults showed lower VA compared to young adults at baseline (*p* < 0.05), as did Simoneau and co-workers (*p* < 0.05) [72]. Similarly, Hvid and co-workers [71] demonstrated that both MVIC and VA were lowered for older adult males compared to young males (*p* < 0.05). Another study, by Knight and co-workers [73], showed an increase in VA (*p* < 0.01) and MVIC (*p* < 0.05) for both older and young adults following a 6-week training protocol at 85% of MVIC (*p* < 0.05). However, while VA reached similar levels between young and older adults following the 6-week RT intervention, older adults were ~30% weaker (*p* < 0.05). In addition, a later study by Hvid and co-workers [70], evaluated the effects of a 12-week RT protocol, at 80% MVIC on VA, muscle mass and gait speed between older active and sedentary adults. Their results showed greater increases in VA, gait speed and MVIC torque for older active adults compared to sedentary older controls (*p* < 0.05). However, these increases in functional performance and VA did not coincide with any significant increases in muscle thickness (*p* > 0.05). Hvid and co-workers also reported a significant association between VA and gait speed (r = 0.67, *p* < 0.05), and argued that improvements in VA were related to increased gait speed and physical performance. Another study, by Simoneau and co-workers [72], evaluated the effect of a twice-a-week PF and DF RT protocol for 6 months using 55–85% MVIC on contractile properties of CT and half relaxation time (1/2 RT), MVIC torque and VA. At baseline, both older active adult subjects and older sedentary adult controls demonstrated an decreased ability to voluntarily activate the PF (*p* = 0.004). Following RT, older adult subjects increased MVIC during PF (*p* = 0.001) and DF (*p* = 0.009) compared to older sedentary controls. However, the authors did not find changes in M-wave amplitude, CT and 1/2 RT for either group, following the intervention (*p* > 0.05). Conversely, VA (*p* < 0.01) and RMS amplitude (*p* < 0.05) of the PF only increased for older adult subjects. Simoneau and co-workers [72] highlighted the fact that the increases in MVIC torque was associated with both the baseline and increases in VA during PF (r = −0.81, *p* = 0.006) and (r = −0.69, *p* = 0.027), respectively. Recently, in a study by Orsatto and co-workers [74], evaluated MVIC torque, sEMG RMS amplitude, contractile properties and VA in resistance-trained young and older adults and untrained older and young adults. In addition to greater RMS amplitude (*p* < 0.001), voluntary maximal explosive torque was greater in both absolute (*p* < 0.001) and normalized values (*p* < 0.03) at 50, 100 and 150 ms for resistance-trained older adults compared to both young and older untrained adults. Furthermore, Orsatto and co-workers demonstrated that resistance-trained older adults, compared to young and older untrained adults displayed a higher ratio between MVIC torque and evoked torque at 50 ms during explosive MVIC, when normalized against evoked torque at 50 ms of knee extension MVIC (*p* < 0.001).

Moreover, two studies [71,75] evaluated the effect of a short-term immobilization and retraining protocol on MVIC, VA and contractile properties. Suetta and co-workers utilized a 4-week unilateral RT protocol of the knee extensors and flexors following a 2-week unilateral immobilization protocol of the same leg. At baseline, both groups showed no differences in VA (*p* > 0.05), or CT (*p* > 0.05). However, at baseline, the peak evoked twitch torque and evoked twitch RFD were reduced in older men compared to young (*p* < 0.05). Following immobilization, both young and older adults showed reduced knee extension MVIC (*p* < 0.05) and evoked twitch RFD (0–30 ms) (*p* < 0.05) and increased CT (*p* < 0.05). While VA remained unchanged for young adults, older adults showed a decrease in VA (*p* < 0.05). Following 4-week RT, both groups regained baseline MVIC (*p*<0.05). Additionally, older adults’ VA returned to baseline (*p* < 0.05), while young adults exceeded their baseline values. In addition, CT and evoked twitch RFD improved for both groups (*p* < 0.05), but to greater extent in older subjects. Of note is the fact that muscle mass decreased more for younger adults than older, following immobilization (*p* < 0.05) and young men showed a greater increase in muscle mass (i.e., retraining) following RT compared to older (*p* < 0.05). The authors argued that older adults regained baseline strength mainly through neural adaptations, compared to younger adults. Interestingly, in a similar study, Hvid and co-workers [71] evaluated the effect of a single high-intensity RT bout on MVIC, VA, contractile properties and cross-over education, following a short-term immobilization protocol of four days. Following the immobilization protocol, MVIC decreased for both older and young adults (*p* < 0.05). While young adults showed no decreases in VA (*p* > 0.05), older adults showed a decreased VA (*p* < 0.05). Following a single session of RT, all parameters returned to baseline for both young and older adults (*p* < 0.05), except MVIC, which did not recover in older adults (*p* > 0.05). For the control leg, no changes were seen in any of the parameters for both older and young, except a decrease in VA in older adults (*p*< 0.05). Finally, a study by Walker and Häkkinen [39] produced noteworthy findings following a 10-week RT protocol for young and older males. The authors evaluated MVIC, VA and appendicular muscle mass. Following the intervention, both groups increased 1-RM leg press performance (*p* < 0.001). Moreover, only older men showed increased VA (*p* < 0.05) and RMS amplitude (*p* < 0.01) following RT. Conversely, only young men showed increased lower-limb lean mass (*p* < 0.01). The increased 1RM performance and lower-limb lean mass were associated in young men only (r = 0.524, *p* = 0.01). The authors argued that while a high-volume “hypertrophic” RT may induce similar improvements in maximal isometric force output for both young and older men, it appears to be due to different mechanisms, being largely hypertrophic for younger men and neural for older men [39].

## 4. Discussion

We carried out a comprehensive systematic review aimed at outlining where and to what extent neural adaptations occur along the descending neuroaxis in response to resistance training in older adults. Overall, our main findings were the following:

(I) At the cortical level, we found that RT training mainly improved corticomotor output in older adults by decreasing intracortical inhibition and shortening CSP, possibly contributing to the observed increases of MVIC [47,48,49], in contrast to previous studies showing increased maximal MEP amplitude in young adults following resistance training [40]. Corticomotor inhibition, is commonly evaluated by administering a supramaximal TMS pulse towards the M1 during an ongoing voluntary isometric contraction, producing a short suppression of sEMG activity, and is termed a cortical silent period (CSP). It has been argued that CSPs are affected by inhibitory γ-aminobutyric acid (GABA), decreasing corticomotor activity [76]. Furthermore, TMS has the ability to evaluate both excitatory and inhibitory cortical pathways [77] and several TMS parameters could possibly provide insight into cortical adaptations following RT. Considering intrinsic excitability, parameters such as resting motor-threshold (rMT) represents the minimum excitability threshold using an oscillating magnetic field [78], expressed as the TMS-stimulator intensity (%) at the moment of depolarization. Submaximal and maximal MEP amplitudes at a given stimulator intensity can shed light on the M1 ability to facilitate descending neural drive. Moreover, distinct silent periods can lend further insight into supraspinal excitability. For example, LICI is produced by administering a 50–150 ms supramaximal stimulus prior to a supramaximal stimulus, termed paired-pulse TMS, reducing the elicited MEP compared to a single stimulus alone [76]. Pharmacological studies have suggested that LICI and CSP are mainly mediated by GABA_b_ [79,80]. This may influence inward rectification and decrease the sensitivity of voltage-gated sodium channels, thereby increasing depolarization thresholds at the level of the cortex and spine [79,80]. Similarly, SICI elicited using a supramaximal stimulus and an inter-stimulus interval of 1–6 ms is instead thought to be mediated at a cortical level (M1) by GABA_a_ [81,82]. Recently, it has been suggested that LICI is affected by both cortical and spinal inhibitory mechanisms [83], making it possible to discern different aspects of cortical inhibition using paired pulse TMS. At rest, older adults showed decreases in relative strength and MEP amplitude compared to young adults [84], alongside prolonged CSP durations, indicating lowered corticomotor excitation. Furthermore, older adults exhibited increased SICI and LICI compared to young adults, indicating increased GABA-mediated cortical inhibition [32]. Thus, decreases in LICI in the absence of SICI may indicate that RT induces neural adaptations at both cortical and spinal levels. The latter may be argued as LICI is known to be mainly influenced by GABA_b_, acting on corticospinal centers, as opposed to GABA_a_ selective corticomotor inhibition. However, a lowering of the CSP in the absence of changes in H-max may instead indicate that this is due to decreased corticomotor inhibition, as shown by Kamen and co-workers. This disinhibition may be mediated by decreases in GABAb, affecting inward rectification, increasing sensitivity of voltage-gated sodium channels and thereby decreasing the depolarization threshold. Otieno and co-workers [49] similarly argued that LICI reductions in older adults post fatiguing exercise may indicate age-related decrease in GABA_b_-mediated LICI inhibition, and that RT acutely provides a neuroplastic stimulus for both cortical and spinal areas. Potentially, factors such as systemic low-grade inflammation in older adults could influence the inhibition of corticospinal areas, as pro-inflammatory molecules such as interleukin-6 can modify synaptic transmission by increasing the expression of GABA_b_ by III/IV-afferent excitation [85].

(II) At a corticospinal track level, several studies demonstrated parallel increases in VA and MVIC achieved in the presence of an increased V-wave amplitude [52,53,54,55]. One may argue that when this is considered in isolation this would indicate increased excitability at both cortical and spinal levels. However, discriminating the relative neuroadaptive influence of RT using H-wave and V-wave is methodologically linked to amplitude cancelation [86]. During the recording of H-max, as the stimulation intensity of Ia afferents increases, collisions between endogenous spinal orthodromic signals and exogenous antidromical signals will cause a state of amplitude cancelation, i.e., if both signals are of equal magnitude, in accordance with Newton’s first law of motion [15]. Thus, maximal H-wave amplitudes will coincide with the highest stimulation intensity immediately prior to the disappearance of the H-wave. The V-wave technique, however, involves recording the magnitude of the reappearance of the H-wave during an MVIC following a complete signal cancelation, and may be appropriate in evaluating the total descending neural drive to the motor neuron pool [19]. The V-wave, hence, represents the ability of a descending neural drive from cortical levels to overcome the state of amplitude cancelation, making the H-wave re-appear [15]. It has been argued that the V-wave represents the total descending corticospinal drive carried to the motor neuron pool, and not only spinal excitability [19]. This is because H-reflex amplitude is mainly influenced by pre-synaptic inhibition from Ia and Ib afferent signals, as well as intrinsic spinal excitability, but is not affected by supraspinal influence [39,56]. V-wave amplitude, therefore, can be altered by supraspinal and spinal excitability [19] and decreased Ia and Ib-afferent activity [11,83]. In the present review, within all studies, increases in V-wave amplitude were achieved in the absence of H-wave alterations, indicating that improved VA and MVIC were mainly due to improved descending cortical drive or increased peripheral excitability [55]. Similarly, Unhjem and co-workers [53] argued that detrimental changes in cortical excitation may play a major role in age-related strength reduction, and that pre-existing motor impairment in older adults could be improved by RT. Therefore, considering that GABA*_b_* inhibits corticospinal excitability and raises the depolarization threshold of MUs, a future research aim may be to evaluate the temporal relationship between the latter parameters and to discover whether these adaptations occur in a top-down manner for older adults.

(III) EMG records not only the electrical activity of the muscle during depolarization, but also the conveyed descending neural output acting at the level of the excitation–contraction coupling, sometimes referred to as neural drive [22,87]. This makes the net EMG activity strongly associated with the magnitude of the descending neural drive to the available motor neuron pool [21]. Commonly extracted EMG parameters include those of amplitude using the average rectified value (ARV) or the root-mean-square (RMS) as a measure of motor neuron pool recruitment. Moreover, frequency parameters such as mean power frequency (MPF) using algorithms such as fast Fourier transform (FFT) can be used in the context of muscle activation in older adults, depicting aspects such as muscle fiber phenotype [88]. Along these lines, it has been reported that higher MPF values correspond to an increased proportion of fast-twitch fibers [89] or the preferential activation of HT-MU [90]. Several studies included in the present review showed an increase in descending neural drive originating from supraspinal areas, depicted by increased RMS amplitude and MVIC for both young and older adults, following RT. These improvements were in general greater for older adults, albeit in the absence of any changes in MPF values for both young and older adults [57,58,60,61,91]. Furthermore, decreases in antagonist co-activation, shown by an inverse relationship of agonist—antagonist RMS amplitude may indicate that a larger portion of neural drive is made available during agonist action for older adults following RT. De Boer and co-workers noted that the above could represent a training-induced accommodation of Ia afferent activity, possibly due to a chronic mechanical load of muscle spindles. Therefore, an accommodation of Ia afferent activity may positively impact reciprocal and recurrent inhibition, increasing agonist force output as a functional consequence. Specifically, a reduction in recurrent inhibition may improve the activation of HT-MUs, requiring greater excitation. The noted changes in “early” and “late” maximal RMS amplitude by Macaluso and co-workers [58] may be indicative of increased recruitment of the available motor neuron pool. The neuron pool in older adults consists, largely, of LT-MUs that exhibit slower contractile properties and lower peak values than HT-MUs. Hence, we can speculate that the latter may explain the selective increase in early RMS values for young compared to older adults.

(IV) For peripheral excitability, older adults have shown decreased maximal MUDR at baseline compared to young adults [38,92], athwart several muscle groups [93]. However, cross-sectional studies have indicated that MUDRs are greater in older weightlifters than age- matched sedentary controls [94]. Moreover, in young adults, it has been demonstrated that a 6-week RT protocol improved MUDR variability and force steadiness [93]. This may indicate that MUDR behavior at the level of the ECC represent one of the final common denominators of adaptational sites along the descending neuroaxis. Several studies demonstrated that older adults exhibited lower maximal MUDR compared to young at baseline [66,67,68,69], which improved following RT for both young and older adults, albeit to a greater extent for older adults [66,67,68,69]. One study indicated that maximal MUDR acutely increased and returned to baseline, coinciding with an increase in contralateral cross-over “education” of the adductor policis [66]. Similarly, another study showed increased maximal MUDR of the abductor digiti minimi and a return to baseline during the second week of RT, also indicating a two-part response, consisting of early disinhibition followed by altered MU-activation [67]. Commonly, depolarization of a MU occurs once synaptic input exceeds the sarcolemmal depolarization threshold (~15 mV). Moreover, in normal conditions, the generated action potential of a muscle should stand in direct proportion to the neural input. However, as postulated by Heckman and co-workers [95], the ionotropic effect of the neuromodulatory system can influence contractile properties by altering the permeability of ion channels. Furthermore, the presence of neuromodulators such as noradrenaline, serotonin and 5-hydroxytryptamine (5HT) can lower the excitability threshold of voltage-sensitive channels [95]. Furthermore, they may also increase persistently inward currents (PICs) and thereby increase and prolong incoming synaptic input [95].

Thus, we may argue that neuromodulation may explain the findings of Griffin [67] and Patten and co-workers [66] in relation to Hebb’s postulate. The latter states that when an axon of cell ‘A’ is in the proximity of the exciting cell ‘B’ or otherwise repeatedly and/or persistently takes part in the firing of it, either a growth process or metabolic adaptation will occur in either or both cells. Thus, when cell ‘A’s efficiency increases, so does cell ‘B’s [96,97]. It is known that GABA*_b_* increases the excitation thresholds not only at a cortical level, but also at the sarcolemma [76,80]. Thus, it is possible that cortical adaptations initially increase MUDR, followed by a reduction in MUDR as the demand for higher discharge rates decreases, due to increased efficiency caused by the neuromodular influence. We argue that this would result in a lowered excitation threshold (i.e., improved efficiency), supporting the two-part adaptation pattern. Moreover, increases in maximal MUDR were seen in parallel with decreased ITT amplitude for both young and older adults, meaning that the evoked amplitude during an MVIC was lowered, indicating greater levels of VA [69]. One possible interpretation may be that these adaptations indicate that higher discharge rates are initially required to activate properly HT-MU prior to improved efficiency, as discussed previously.

Knight and co-workers [69] indicated the link between MUDR and VA. For both young and older adults, improvements in maximal MUDR were strongly associated with decreased ITT-amplitude, indicating a decrease in the neural deficit. For associations of MUDR and CAR, significant associations were only found for older adults. CAR has been shown to be less sensitive to changes than ITT, possibly indicating that functional improvements in force are dependent on MUDR in older adults [98]. Moreover, studies evaluating the effect of RT on MU synchronization and variability did not detect any improvements in MVIC and MU synchronization. Conversely, force variability decreased at lower intensities for older adults, possibly indicating that low-load RT improves force steadiness by reducing motor unit firing variability, rather than by changing motor unit synchronization. The latter seems to be in accordance with the previous literature, which has questioned the role of MU synchronization on VA and force production [23].

(V) The majority of the studies showed that older adults exhibit lower VA and MVIC at baseline compared to young adults [39,70,71,72,73]. However, cross-sectional findings indicate a maintained contractile quality and peripheral excitability in older resistance-trained men [74]. Moreover, several studies indicated increased physical performance following RT that coincided with improved VA, shown by a decreased ITT amplitude. For example, Hvid and co-workers reported significant associations between VA and gait speed, arguing that improvements in VA are related to increased gait speed and physical performance [70]. Furthermore, several studies indicated that improvements in MVIC and performance parameters occurred in the absence of increased muscle mass, and, conversely, in the presence of improved VA [39,57,70]. Additionally, studies implementing lower-limb immobilization and re-training, indicated a susceptibility to VA losses in older adults, depicted by greater decreases in VA for older adults compared to young [71,75]. However, following retraining, both young and older adults successfully regained baseline VA [71,75]. Hvid and co-workers noted, however, that VA of the non-immobilized leg decreased only in older adults, possibly indicating that an adequate descending neural drive may be required to maintain bilateral VA. However, whether improved contractile properties occur alongside increased VA is less certain. While decreases in CT following RT occurred, the evoked force remained unchanged. Simoneau and co-workers [72] argued that increased MVIC torque, in the absence of altered contractile properties in older adults, was likely explained by an increased descending neural drive. Lastly, a study by Walker and Häkkinen [39] utilizing a moderate-load RT protocol, demonstrated that young and older adults exhibit similar increases in MVIC albeit through different adaptation mechanisms, these being largely hypertrophic in nature for the young in the absence of neural adaptations, and conversely, mainly neural, in the absence of skeletal muscle hypertrophy, for older adults. However, when interpreting the above findings, it is important to take into consideration the impact of possible methodological issues in the evaluation of VA, as the standard ITT protocol, using a single 1-ms square wave has been shown to be able to reliably elicit an interpolated twitch [13,99]. The use of 1 ms square waves of doublets to octets stimulation has been suggested to improve the detection of VA, since multiple twitches evoke larger interpolated twitch responses in larger muscles [99]. However, the studies included in this review evaluated VA using 1 ms single-to-octet stimulation trains.

We argued that, given the premise that the descending neural drive to the muscle precedes the onset of skeletal muscle contractions, it might be plausible that any preceding impairments along the descending neuroaxis would thereby impair the VA of skeletal muscles. Overall, the studies outlined in the present review point towards an increased descending neural drive following RT for older adults, resulting from decreased intracortical inhibition. The latter was depicted by increased V-wave amplitude and decreased LICI, in the absence of MEP max, M-max and H-wave changes. Therefore, increased corticomotor excitability may explain the greater EMG increases described in this review, considering that RMS amplitude is partially affected by the magnitude of the descending neural drive, generally seen alongside increased MVIC. Increases in maximal MUDR were preferentially improved during higher intensities (MVIC), but not lower intensities. Furthermore, as MUDR increased, ITT amplitude subsequently decreased (i.e., increased VA), which may point towards an increased activation of HT-MUs [72]. This decrease in ITT amplitude may represent cortical adaptations, indicated by an increased V-wave amplitude in the absence of H-wave alterations. In addition, neuromodular adaptations, prompted by noradrenaline, serotonin and 5HT, may increase the efficiency of persistently inward currents, thus prolonging the synaptic input of MUs [96,97]. Lastly, overall VA increased largely in older adults compared to young following RT, increasing physical performance in older adults. Therefore, it may be plausible to argue for a chain of adaptations, where superior adaptations of the descending neuroaxis precede those of inferior levels.

## 5. Future Directions

In the context of age-related decline of muscle mass and function, it is of great importance to differentiate between the intrinsic contractile force of a muscle and the ability to activate muscles voluntarily. The above-discussed findings indicate that superior impairments in the descending neuroaxis affect the neural output of lower levels, and likely the functional capacity. Considering that VA losses are two-to-five-times greater than those of skeletal muscle mass [2,5,6], it may be possible that impairments along the descending neuroaxis precede losses in muscle mass. Furthermore, this raises the question whether losses of neuromuscular function with preserved muscle mass should be addressed in the same manner as sarcopenia, as noted by Clark and Manini [2,10]. This may present important implications for the treatment of the two separate conditions, given that the pathophysiological mechanisms may not be equal. The latter indicates the necessity for longitudinal studies evaluating the impact of RT on VA in older adults. In the present review, studies indicate that older adults generally display attenuated levels of excitation and increased inhibition along the descending neuroaxis, and that RT favors neural adaptations rather than morphological adaptations in older adults, compared to young. Moreover, superior impairments appear to affect the neural output of lower levels, and this is supported by similar observations in dynapenic patients displaying normal muscle mass alongside decreased corticomotor excitation, VA, and functional capacity [84]. Furthermore, when the previously mentioned techniques are used in isolation, discerning the relative neuroadaptive contribution of each level may be difficult. We therefore highlight the need for standardization in the use of combined techniques in the evaluation of neural adaptations along the descending neuroaxis and their relative influence on VA. We argue that the choice of technique used for each level should be in relation to a clearly stated hypothesis, to explicitly state: (a) the relative neuroadaptive contribution of each level following a stimulus, acutely or over time, and (b) its effect on VA of muscle and functional capacity. We outline below a model based on tables I-V, which may aid the appropriate choice of technique in the evaluation of separate levels of the descending neuroaxis and the corresponding relative neuroadaptive contribution following resistance training (Figure 2).

## 6. Conclusions

The neural contribution to adequate neuromuscular function and VA of skeletal muscles should not be understated. It appears that young and older adults adapt differently in response to resistance training, favoring skeletal muscle hypertrophy in the young and neural adaptations in older adults. We argue that improved understanding of the relative neuroadaptive contribution along the descending neuroaxis in response to RT may contribute to the formulation of non-pharmacological interventions, such as training protocols, which can improve physical function and life quality for older adults. Furthermore, while the literature is clear that a plethora of neural adaptations improve VA and physical performance in older adults, even in the absence of increased muscle mass, there is a lack of a standardized systematic evaluation of the relative neuroadaptive contribution to VA following RT. We therefore encourage future studies to systematically implement techniques and parameters used in such an evaluation, signifying where and to what extent along the descending neuroaxis these adaptations occur.

## Figures and Tables

**Figure 1 brainsci-13-00679-f001:**
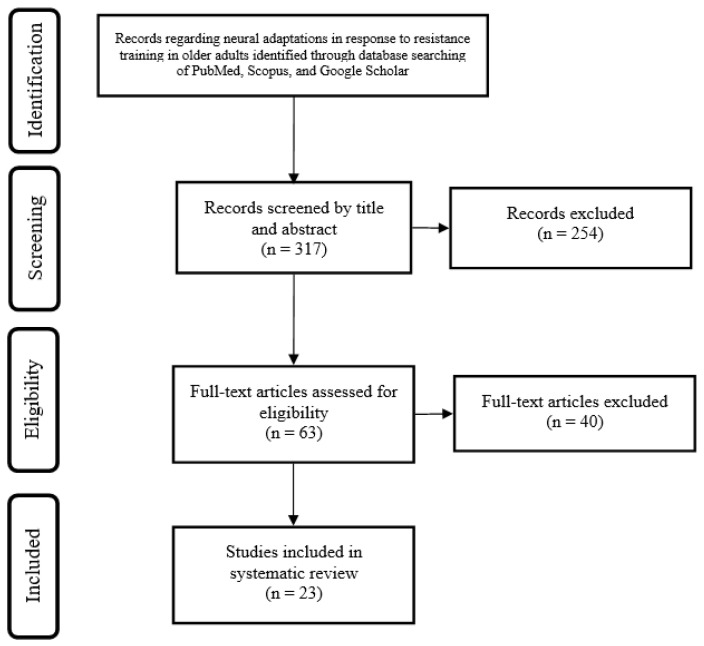
Flow-chart illustration of the literature search.

**Figure 2 brainsci-13-00679-f002:**
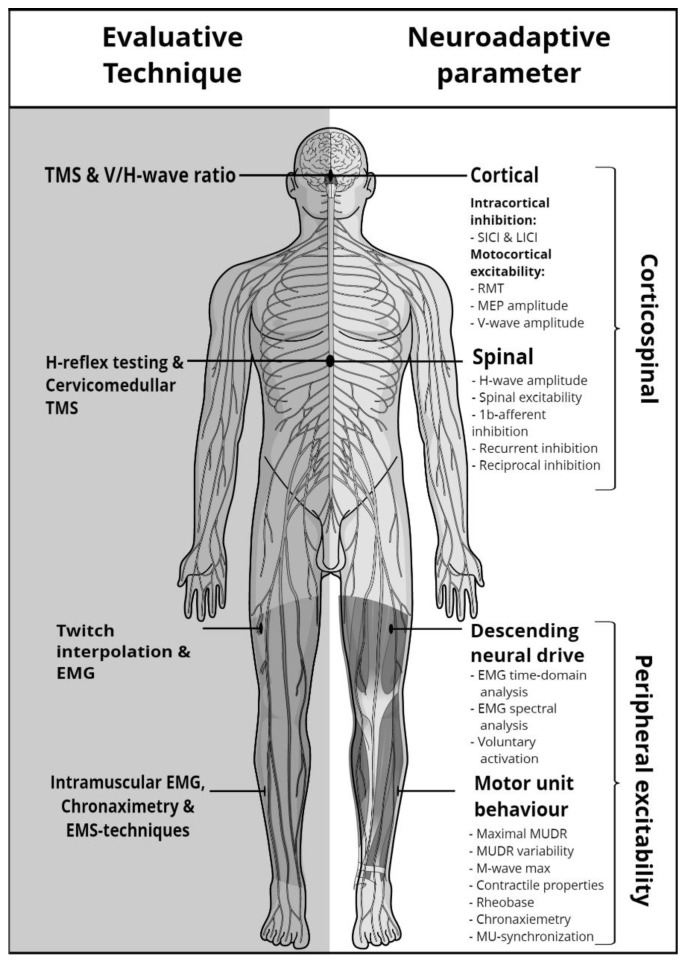
Potential sites of evaluations along the descending neuroaxis with corresponding techniques and parameters. Legend: TMS, Transcranial magnetic stimulation; V-wave, Volitional wave; H-wave, Hoffman wave; H-reflex, Hoffman reflex; EMG, Electromyography; EMS, Electrical muscle stimulation; SICI, Short-interval intracortical inhibition; LICI, Long-interval, Intracortical inhibition; RMT, Resting motor threshold; MEP, Motor-evoked potential; MUDR, Motor-unit discharge rate; M-max, Maximal elicited muscle amplitude.

**Table 1 brainsci-13-00679-t001:** Cortical-level studies evaluating supraspinal adaptations using TMS following RT in older adults.

Study	Subjects (n) and Age (Years)	Controls (n) and Age (Years)	Intervention and Duration	Aspects Evaluated and Methodology	Main Findings
Kamen, 2014 [47]	30 older adults (72.9 ± 4.6)	30 young adults (21.9 ± 3.1)	2-week RT protocol of the dorsi flexor muscles (3×/week at 85% MVIC)	Single-pulse TMS CSP MEP max MVIC force H-reflex and M-wave amplitude	Older adults showed lower MVIC, M-wave amplitude, and prolonged CSP at BL. MVIC increased and CSP decreased more in old adults compared to young, while MEP max and H-wave amp. remained unchanged. Improvements were overall greater in older than young.
Penzer et al., 2015 [48]	10 older adults (71.4 ± 6.3)	8 older adults (71.4 ± 6.4)	6-week RT protocol (3×/week at 85% 1RM) or RT + Balance training (6 weeks, 3×/week at 85% MVIC)	Single-pulse TMS: MEP max CAR H-wave max amplitude H-wave 50% stimulus threshold	No between-group BL differences for any parameter. MVIC, CAR increased, and 50% H-max threshold decreased more in the RT group, while MEP max and M-max remained unchanged in both groups.
Otieno et al., 2021 [49]	18 older adults (69 ± 5)	19 young adults (23 ± 4)	Single bout 15 min isometric RT using 25% of MVIC sEMG amplitude and 10 × 2 min of 25% MVIC sEMG amplitude.	Paired and single-pulse TMS: SICI (2 ms ISI) LICI (100 ms ISI)	BL differences were shown for all measures between young and old adults. SICI remained unchanged in both groups, while LICI decreased for only older adults acutely following RT.

Legend: BT, Balance training; BL, Baseline; TMS, Transcranial magnetic stimulation; CAR, Central activation ratio; H-Wave, Hoffman-reflex wave; MEP, Motor-evoked potential; MVIC, Maximum voluntary isometric contraction; CSP, Cortical silent period; M-Wave, elicited muscle amplitude; ISI, Inter-stimulus interval; sEMG, surface electromyography; SICI, Short-interval intracortical inhibition; LICI, Long-interval intracortical inhibition; VA, Voluntary activation.

**Table 2 brainsci-13-00679-t002:** Corticospinal-level studies evaluating corticospinal adaptations using TMS and V-wave methodologies following RT in older adults.

Study	Subjects (n) and Age (Years)	Controls (n) and Age (Years)	Intervention and Duration	Aspects Evaluated and Methodology	Main Findings
Unhjelm et al., 2015 [52]	9 older adults (74 ± 6)	8 young adult male (24 ± 4)	8-week knee extensor RT protocol (3×/week at 85% MVIC)	Ultrasound muscle thickness MVIC force H/M-ratio V/M-ratio	At BL older adults showed attenuation of V/M ratio and prolonged H-wave latency, but no difference in H/M ratio compared to young adult controls. RT improved V/M-ratio and MVIC in older adults, but not the H/M-ratio compared to young adult controls.
Unhjem et al., 2021 [53]	36 older adults (73 ± 4)	30 young adults (21.9 ± 3.1)	3-week of plantar flexion RT protocol of: MST (90% MVIC) or UBT	V/M ratio MVIC force ITT	The MST group increased MVIC and RFD. While the UBT did not display improvements of the same parameters. For MST parallel improvements of Vmax/M-max ratio, MVIC was shown. A tendency for increased VA was only shown in the MST groups. However, no changes were observed in H-max/M-max ratio for any group.
Tøien et al., 2018 [54]	11 older adults (73 ± 4)	12 older adults (73 ± 4)	3-week RT protocol (3×/week at 85% MVIC)	MVIC force H/M V/M-ratio	Older subjects showed parallel increases of ipsilateral and contralateral MVIC, and V/M-ratio compared to older controls. Increased V/M ratio was strongly associated with improved VA. However, no changes were observed in H/M ratio after RT.
Scaglione et al., 2002 [55]	14 older adults (65–80)	10 young adults (24–35)	16-week of plantar flexor RT protocol (3×/week at 50–80% of MVIC)	H/M- ratio ITT CT and ½ RT	Following RT older adults showed greater VA improvements compared to young adults, while H-wave amplitude remained unchanged in both groups. MVIC increased alongside VA. CT decreased and ½ RT remained unchanged.

Legend: RT, Resistance training; MVIC, Maximum voluntary isometric contraction; M-Wave, elicited muscle amplitude; sEMG, surface electromyography; VA, Voluntary activation; H/M ratio, H-wave-M-max ratio; V/M ratio, V-wave-M-max ratio; MST, Maximal strength training; UBT, Unloaded ballistic training; ITT, Interpolated twitch.

**Table 3 brainsci-13-00679-t003:** Resistance training studies evaluating EMG parameters following RT in older adults.

Study	Subjects (n) and Age (Years)	Controls (n) and Age (Years)	Intervention and Duration	Aspects Evaluated and Methodology	Main Findings
Suetta et al., 2004 [57]	RT: 18 older adults (71 ± 12.5)	EMS: 9 older adults (69 ± 7.5) Standard rehab: 9 older adults (69 ± 8)	12-week rehab using: RT (3×/wk) EMS or Standard rehab	MVIC force sEMG RMS amplitude Evoked RFD and impulse	Old adults (RT) improved MVIC, evoked RFD and impulse and no change was seen for the other groups. RMS amplitude increased for all group, but to a greater extent in the RT group.
Macaluso et al., 2000 [58]	RT: 8 older women (70–79) RT: 8 young women (18–30)	8 older women (70–79) 8 young women (18–30)	6-week RT protocol (3×/week 40–80% MVIC)	MVIC force RMS amplitude (Early and late) MPF (Early and late)	Early RMS increased for young, but not old. Conversely, late RMS increased for old, but not young following training. MVIC increased for both groups, while MPF remained unchanged for both groups.
Häkkinen et al., 2001 [59]	11 older men (72 ± 3) 10 older women (67 ± 3)	10 middle-aged men (42 ± 2) 11 middle-aged women (39 ± 3)	6-month RT-protocol (2×/week 50–80% MVIC)	MVIC force Agonist-antagonist co-activation: sEMG RMS amplitude	MVIC and VL RMS amplitude increased in both young and older adults, with no between-group differences. Antagonist BF activity during the isometric knee extension remained unaltered in controls following RT. However, for older women antagonist coactivation decreased.
De Boer et al., 2007 [60]	12 older adults (74.2 ± 3.1)	8 older adults (73.6 ± 4.3)	52-week PF RT protocol (3×/week 60–80% MVIC)	PF and DF MVIC force and RMS amplitude at 20, 10° (DF) and 0, 10, 20, 30° (PF).	Only the RT group improved MVIC of the PF and decreased DF MVIC in parallel with increased agonist amplitude (PF) and decreased antagonist amplitude (DF) for all joint angles.
Cannon et al., 2007 [61]	8 older women (69.8 ± 6.6)	9 young women (25.0 ± 4.0)	10-week RT protocol (3×/week at 75% 1RM)	Peak MVIC torque LCSA ITT torque RMS amplitude	ITT amplitude did not improve after the training period for either group. RMS amplitude increased in both groups, but to a larger degree in old. Young displayed higher RMS amplitude prior and following the intervention. Both groups showed similar increases in LCSA and peak isometric torque, with no between group differences.

Legend: RT, Resistance training; EMS, Electrical muscle stimulation; sEMG, Surface electromyography; RFD, Rate of twitch force development; RMS, Root-mean-square; MPF, Mean power frequency; LCSA, Lean cross-sectional area; MRI, Magnetic resonance imagining; PF, Plantar flexor; DF, Dorsi flexor; BF, Biceps femoris.

**Table 4 brainsci-13-00679-t004:** Peripheral excitability studies evaluating motor-unit discharge rates and synchronization following RT in older adults.

Study	Subjects (n) and Age (Years)	Controls (n) and Age (Years)	Intervention and Duration	Aspects Evaluated and Methodology	Main Findings
Patten et al., 2001 [66]	6 older adults (75.8 ± 7.4)	6 young adults (23.2 ± 3.5)	Unilateral isometric RT of the ADM for 5 days/week for 6 weeks at 85% of MVIC.	VL MUDR MVIC force Cross-over education (ipsilateral and contralateral)	At BL maximal MUDR were lower in older adults compared to young. After RT, MUDR and MVIC increased for both groups, but greater in older adults. Increased contralateral MUDR were only shown in older adults.
Griffin et al., 2009 [67]	10 older adults (66.1 ± 1.27)	9 young adults (28.2 ± 9.5)	4-week low-load RT protocol of 1st FDI and ADP muscle using fixed loads of 10–20 lb.	Max MUDR MU synchronization Force variability (CV) at 2, 4, 8, 12% MVIC force	Only older adults improved max MUDR and CV of the index finger and thumb, and CV at 2, 4% of MVIC. No improvements at 8 and 12% of MVIC. However, MVIC and MU synchronization was unaffected in both groups.
Kamen et al., 2004 [68]	RT: 7 older women (mean age 77)	8 young women (mean age: 21)	6-week RT protocol (3×/week 40–80% MVIC)	MVIC force VL MUDR at 10, 50, 100% MVIC force	BL max MUDR were greater in young than old adults. MVIC increased for both groups, but to a larger degree in old. Max MUDR improved for both groups, but to a larger extent in old. No change in MUDR was seen at 10 or 50% of MVIC.
Knight et al., 2008 [69]	6 older (67–81)	8 young adults (18–29)	6-week knee extensor RT protocol (3×/week 85% MVIC)	MUDR CAR ITT	RT improved all parameters for both groups. Significant association of improved max MUDR and MVIC torque, and for max MUDR and ITT amplitude for both young and old adults. Significant association of max MUDR and CAR was only shown in old.

Legend: RT, Resistance training; EMG, Electromyography; MUDR, Motor-unit discharge rate; CAR, Central activation ratio; MVIC, Maximal voluntary isometric contraction; CV, Coefficient of variation; ADP, adductor pollicis; ADM, Abductor digitorum minimi; FDRI, first dorsal interosseous; ITT, Interpolated twitch technique; BL, Baseline; VL, Vastus lateralis.

**Table 5 brainsci-13-00679-t005:** Studies evaluating voluntary muscle activation using ITT or CAR separately or in combination with the methodologies of I–IV, following RT in older adults.

Study	Subjects (n) and Age (Years)	Controls (n) and Age (Years)	Intervention and Duration	Aspects Evaluated and Methodology	Main Findings
Walker et al., 2014 [39]	26 older men (64 ± 8)	23 young men (29 ± 9)	Young: RT (2×/w, 60–85% 1RM) Old: RT (2×/w, 60–85% 1RM)	ITT sEMG amplitude. Knee extensor MVIC force LCSA	Older men displayed lower BL VA compared to young. Both young and old RT groups improved 1-RM leg press performance. However, increased sEMG amplitude and VA was evident only for older men. Conversely, only young increased lower-limb LCSA. Increased VA and sEMG amplitude were associated with increased 1RM in young men only.
Hvid et al., 2016 [70]	16 older adults (82.3 ± 1.3)	21 older adults (81.6 ± 1.1)	12-week high-load power training (2×/week at 70–80% 1RM)	Muscle thickness Maximal isometric torque ITT amplitude 2-min gait speed	Older subjects increased VA, gait speed and MVIC torque, but no significant increase in muscle thickness was observed. A significant association of improved VA and gait speed was detected.
Hvid et al., 2018 [71]	11 older healthy men (67.2 ± 1.0).	11 young healthy men (24.3 ± 0.9)	Single high-intensity RT session following a unilateral disuse protocol (4 days, knee brace)	MVIC force ITT Maximal evoked muscle twitch force	Baseline MVIC and VA was lower for older adults compared to young adults. Following immobilization, MVIC decreased for both groups, but to a larger extent in old, and no decrease in VA was seen in young but decreased for old adults. After RT all parameters regained baseline levels for both groups, except MVIC for older adults. For the control leg and no changes were seen in either group, except decrease in voluntary activation for older adults.
Simoneau et al., 2007 [72]	11 older adults (78.1 ± 3.1)	9 older adults (75.9 ± 3.4)	6-month RT protocol (2×/week 55–85% of 1RM)	MVIC torque M-wave amplitude CT & ½-RT ITT amplitude	Older subjects increased PF and DF MVIC of after RT compared to sedentary controls. CT and 1/2 RT did not change in either group. VA increased for the PF compared to controls. RT group increased VA while controls did not. MVIC torque was associated with improved VA at both BL and after RT.
Knight et al., 2001 [73]	6 older (67–81)	8 young adults (18–29)	6-week knee extensor RT protocol (3×/week 85% 1RM)	MVIC torque CAR	Older adults had lower BL VA compared to young. VA and MVIC increased for both groups, but to a larger degree in older. However, older adults were ~30% weaker than young despite displaying similar VA levels in absolute terms.
Orssatto et al., 2020 [74]	12 older trained adults (63.6 ± 3.8) 14 young, trained adults (26.7 ± 3.4)	14 older untrained adults (65.6 ± 2.9) 14 young untrained adults (26.2 ± 3.7)	Cross-sectional study	Knee extension MVIC explosive torque 50–150 ms Maximal sEMG amplitude 50 ms voluntary explosive and evoked torque ratio	For trained young and old adults, MVIC torque and sEMG amplitude at 50, 100, 150 ms was higher both absolute and normalized compared to untrained. Older and young trained adult showed higher MVIC torque and evoked torque ratio at 50 ms. However, younger trained adults displayed higher evoked torque at 50 ms compared to old adults, but not compared to older trained adults.
Suetta et al., 2009 [75]	9 older men (67.3 ± 9.2)	11 young men (24.4 ± 4.2)	4-week unilateral RT protocol of the knee extensors and flexors (3×/week 1, at 10–12 RM) following a 2-week immobilization of the same leg	MVIC force Muscle mass: DEXA Contractile properties: CT CAR (doublet 0.1 ms square waves)	BL VA, and CT were not different between groups. Post-immobilization, resting twitch torque and RFD were reduced for old only. Young and old adults decreased MVIC, evoked RFD and torque. Only older adults decreased VA. After RT VA returned to BL for old and young surpassed BL VA values. CT decreased for both groups, while twitch RFD and torque improved more in older than young adults, and muscle mass increased to a larger extent in young adults.

Legend: RT, Resistance training; EMG, Electromyography; MVIC, Maximal voluntary isometric contraction; CT, Contraction time; ½-RT, Half-relaxation time; ITT, Interpolated twitch technique; DEXA, Dual energy x-ray absorptiometry; CAR, Central activation ratio; QF, Quadriceps femoris.

## Data Availability

The complete data are accessible upon request from the author.

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
