# Peer review of "Relative Neuroadaptive Effect of Resistance Training along the Descending Neuroaxis in Older Adults"

_brainsci, 2023, doi:10.3390/brainsci13040679_

Round 1

Reviewer 1 Report

The purpose of this review article was to review the relative contribution of neural adaptations to increases in strength following resistance training (RT) in older adults. More specifically, emphasis was placed on the sites within the CNS that contribute to increases in strength in older adults as well as the techniques used to quantify these effects. The major conclusions of the review were: 1) that changes in neural drive from the cortex (less excitation, greater inhibition) negatively influence adaptations to resistance training at lower levels in the nervous system and likely the muscle; and 2) older adults display greater neural changes following RT compared to muscular changes. Thus, interventions may need to focus on neural adaptations in older adults to maintain or slow the rate of decline in muscle strength in older adults.

Overall, the review was very extensive. The Material and Methods section seemed to be rigorous in regard to the criteria for including and excluding studies and related issues. The overall layout and organization of the review was also good and easy to follow.  The figures and tables conveyed a great amount of information, were well done, and complemented the text making the findings of the studies easier to understand. The authors interpretation of all the studies and groups of them also seemed to be adequate and appropriate.

In summary, I really liked this review and I think it adds to the literature by providing a paper that any scientist can go for a solid overview of this topic. Furthermore, I don’t think there has a recent extensive review on this topic and the readers of Brain Sciences will be interested in the topic.

Nonetheless, I have several minor things that should be changed before the paper is published.

1.      Although the paper was very well-written overall, there are some typos, formatting inconsistencies, and other minor errors in the paper. There are too many for me to point out individually. Thus, more proofreading is needed. I will list just some examples below. In addition, I add a few other minor wording issues in some sentences.

a.       Line 259,260,262, 434. The p value reporting of < and = sometimes have spaces before and after and sometime not. Line 434 the P is capital where elsewhere it is not. Check all of this sort of thing throughout and be consistent.

b.      Lines 306 and 309, I think the authors mean Ib and not Iib ? correct

c.       Table II the legend at the bottom has a formatting issue on the left side of the first line.

d.      Lines 596 and 597 I it should read Ia afferent not Ia-afferent

e.       Figure 2 the line on the figure runs through the words “peripheral exciteability” and through “corticospinal” on the right.

f.        Line 51 sentence starting with “Thereby” needs rewording

g.      I don’t think motor unit synchronization plays a role in ability to voluntarily activate muscles or increase muscle force. Original studies in the 70s said this but it is has been known to not be true for awhile. I also believe this recent study https://pubmed.ncbi.nlm.nih.gov/31120812/ has put this and related issues to rest. Thus, I recommend rewording that sentence.

h.      Line 111, why use & and not the word and ?

i.        Line 165 appears to have more spaces after the period than other sentences, this happens other times in the paper like perhaps line 637.

j.        The term motocortical in line 554 I have never seen used before, corticomotor has been obviously. Would suggest rewording in some way.

k.      Bibliography needs a lot of checking for instance line 954 journal title is abbreviated other plascs it is not such as line 959. Many instances of this inconsistency.

Author Response

Thank you for taking the time to review our manuscript entitled “Relative neuroadaptive effect of resistance training along the descending neuroaxis in older adults”. We appreciate taking the time to share your expertise and valuable feedback on our work. We hope that our research will be of interest to the readers of MDPI Brain Sciences, and that the implementation of your comments will help us improve the quality of our manuscript according to the standard of MDPI Brain Sciences.

On behalf of the authors, we thank you for this opportunity.

Reviewer 1 – Comments and answers

  1. Although the paper was very well-written overall, there are some typos, formatting inconsistencies, and other minor errors in the paper. There are too many for me to point out individually. Thus, more proofreading is needed. I will list just some examples below. In addition, I add a few other minor wording issues in some sentences.

Answer: We thank the reviewer for their kind words and insightful comments. Aside from individually addressing the comments outlined below we have carefully tried to address typos, formatting, and miscellaneous errors in the article. A few parts of the text featured heavy editing, making track changes unfeasible. However, we have used track changes in the other changes of the manuscript.

  1. Line 259,260,262, 434. The p value reporting of < and = sometimes have spaces before and after and sometime not. Line 434 the P is capital where elsewhere it is not. Check all of this sort of thing throughout and be consistent.

Answer: We thank the reviewer for pointing this out, this was a mistake on our part which we have now addressed this aspect continually throughout the text.

  1. Lines 306 and 309, I think the authors mean Ib and not Iib ? correct

Answer: Thank you for pointing this out, this is indeed an unfortunate typo that has since been corrected.

  1. Table II the legend at the bottom has a formatting issue on the left side of the first line.

Answer: We have addressed what we believe the reviewer was referring to in Table II.

  1. Lines 596 and 597 I it should read Ia afferent not Ia-afferent

Answer: Thank you, we have changed all instances of Ia-afferent to the correct Ia afferent.

  1. Figure 2 the line on the figure runs through the words “peripheral excitability” and through “corticospinal” on the right.

Answer: We thank the reviewer for carefully pointing this out, this was most likely due to a rendering error in the figure. Additionally, we have added the missing figure caption.

  1. Line 51 sentence starting with “Thereby” needs rewording.

Answer: We have reworded the sentence accordingly.

  1. I don’t think motor unit synchronization plays a role in ability to voluntarily activate muscles or increase muscle force. Original studies in the 70s said this but it is having been known to not be true for a while. I also believe this recent study https://pubmed.ncbi.nlm.nih.gov/31120812/ has put this and related issues to rest. Thus, I recommend rewording that sentence.

We'd like to thank the reviewer for the well thought out comment as well as providing a reference to current literature. The statement has been reworded and we have added above reference in the introduction that highlights this aspect. Furthermore, our findings outlined in Table IV agrees with reviewer's insightful comment, no changes in MU synchronization were observed following RT. This was greatly appreciated.

  1. Line 111, why use & and not the word and?

Answer: We thank the reviewer for pointing out this error on our part, we have removed it and changed to the suggested "and".

  1. Line 165 appears to have more spaces after the period than other sentences, this happens other times in the paper like perhaps line 637.

Answer: Thank you for pointing this out, however in the attached version we have not been able to detect this extra space you are referring to. We will attempt to check with the journal so that there is no issue with the formatting of the article. Thank you!

  1. The term motocortical in line 554 I have never seen used before, corticomotor has been obviously. Would suggest rewording in some way.

Answer: We thank the reviewer for pointing this out. We have changed the instances of motocortical to corticomotor to accommodate this comment. Thank you!

  1. Bibliography needs a lot of checking for instance line 954 journal title is abbreviated other plascs it is not such as line 959. Many instances of this inconsistency.

Answer: Thank you, we have carefully gone through the bibliography for errors such as the one mentioned above, all journal titles should now be abbreviated.

Reviewer 2 Report

The authors provide an extensive review of studies which investigate the effect of resistance training on the sensorimotor system in elderly individuals. The review examined 23 studies that involved several techniques and different populations. I would recommend some revision of the text and presentations of results, before accepting the manuscript for publication. Please, find my comments in the pdf.

Author Response

Thank you for taking the time to review our manuscript entitled “Relative neuroadaptive effect of resistance training along the descending neuroaxis in older adults”. We appreciate taking the time to share your expertise and valuable feedback on our work. We hope that our research will be of interest to the readers of MDPI Brain Sciences, and that the implementation of your comments will help us improve the quality of our manuscript according to the standard of MDPI Brain Sciences.

On behalf of the authors, we thank you for this opportunity.

  1. Is this referring to young adults?

Answer: We thank the reviewer for pointing this out, this is correct, this refers to young adults. We have now changed this.

2-3.  In the text is missing the reference to the formula. Also, I believe the equation is wrong, parenthesis are missing or misplaced... Also here, it would be better to explicitly point at the equation which should be listed with a number.

Answer: We are grateful that the reviewer pointed this out to us, we have now added the title Equation 1 and Equation 2 for each corresponding equation, as well as adding the reference. Equation 1 has been updated to match the one of Gandevia and co-workers 1999 for calculation voluntary activation percentage [13].

  1. I would add a proper definition of resistance training. Can the authors provide an example and/or describe the types of tasks involved?

Answer: We have added a brief definition of resistance training, thank you.

  1. I would explicit the criteria to help the reader understand the process.

Answer: We have added the full PEDro quality assessment criterions, and their corresponding explanations in the appendix, thank you.

  1. There must be a mistake here:

7-6 excellent

<5 (i.e. 4) good

5 moderate

 < 4 (i.e. 3)  poor

I would assume that 5 is good, 4 is moderate and <4 is poor... Please double check and correct.

Answer: We are very grateful for the reviewer to catch this mistake on our part. This is indeed an unfortunate typo. This has now been corrected in the most recent version of the manuscript. Thank you.

  1. What is the appendix? I could not find it. Which one is attachment 1? Please refer properly to the material provided.

Answer: We thank the reviewer for pointing this out. During the submission process of the article, we provided a RAR-file with all the supplementary material, including tables, figures and appendix. However, not all the supplementary material had been inserted into this current manuscript. We have addressed this by including the material in the appendix, in addition to the following questions of the reviewers which pertains to the appendix. Mention of this material is stated within the text in this format "See attachment x of the appendix".

  1. Again, to which file are the authors referring? Also, if these elements are necessary in the paper they should not be in the supplementary material, but maybe a table should be embedded in the main text:

Answer: While this regards the same as question 7 of the reviewer, we simply wish to state that we were asked to provide this as either an appendix or supplementary material. We have now included this within the Appendix A.

  1. Acceptable quality means with score 4 or higher?

Answer: We thank the reviewer for addressing this, we have for the sake of clarity added this explicitly next to this statement.

  1. Is there an effect of age in the RT training results? Did any study explore the differences in longitudinal studies? Please elaborate in the discussion at least

Answer: We are grateful for this insightful comment. We believe this is an interesting viewpoint. While there certainly is literature regarding RT training and its effects in older adults in the relative short term, there's a lack of longitudinal studies overall, and an even greater lack when it comes to studies evaluating neural adaptations. Likely due to practical constraints. Thus, this would be an important aspect for future studies to address.  We address this briefly in the discussion. based on this comment.

  1. Again, here the name of the file does not match the referenced name.

Answer: Thank you for this comment, this is appreciated. This has been addressed in the same manner as the previous comments.

  1. Please, double check whether these points match the definition provided in the methods.

Answer: Again, we thank the reviewer for carefully pointing this out previously. We have carefully confirmed that the scoring outlined here match the definition given in the methods section.

  1. This paragraph, from lines 213 to 250 seems more a discussion rather than a results paragraph. I would first list the results from the included papers and then comment those in the appropriate section. Similarly, all the paragraphs below in each section of the results. It is ok to explain some of the techniques used in the studies, but it should be brief. I would leave this level of detail in the introduction or discussion.

Answer: We thank the reviewer for taking the time to explaining this point to us, and we agree that this is too extensive to be featured in the results. We have attempted to address this by featuring only a brief overview of the technique and physiological interpretation. In addition to implementing these paragraphs to the corresponding section in the discussion.

  1. Young adults?

Answer: This is indeed a typo; we apologize for this and have now addressed this.

  1. Something happened here, maybe when creating the pdf

Answer: Most likely this is a formatting error, in the most recent version of the manuscript this does not appear to be an issue. But please let us know if anything looks strange in the current version, thank you.

  1. This figure is missing a caption

Answer: We thank the reviewer for this comment, we have now added the figure and its original caption, thank you.

  1. How is this result related to the neural cognitive decline of older adults? I wonder whether any study correlated adaptations in controlling skeletal muscles with cognitive scales.

Answer: We appreciate the insightful comment from the reviewer. Originally, we were considering featuring an aspect of the review dedicated to cognition and voluntary activation. There are currently some fascinating studies that evaluate handgrip as a marker for cognitive decline (See [7] Carson and co-workers). However, we decided this may be too far outside the scope of the perspective of the descending neuroaxis. Which is why we decided not to include this aspect. A very intriguing comment regardless.

  1. Again not dure if this match the material I have access to.

Answer: We thank the reviewer for pointing this out. We purposefully left it blank in the submission in case we would have been provided with an URL by the journal for our uploaded supplementary material. However, since we included the appendix, we have removed this.